# UVC-Based Air Disinfection Systems for Rapid Inactivation of SARS-CoV-2 Present in the Air

**DOI:** 10.3390/pathogens12030419

**Published:** 2023-03-07

**Authors:** Harry Garg, Rajesh P. Ringe, Supankar Das, Suraj Parkash, Bhuwaneshwar Thakur, Rathina Delipan, Ajay Kumar, Kishor Kulkarni, Kanika Bansal, Prabhu B. Patil, Tabish Alam, Nagesh Babu Balam, Chandan Swaroop Meena, Krishan Gopal Thakur, Ashok Kumar, Ashwani Kumar

**Affiliations:** 1Central Scientific Instruments Organisation, Council of Scientific and Industrial Research (CSIR), Sector 30C, Chandigarh 160030, India; 2Institute of Microbial Technology, Council of Scientific and Industrial Research (CSIR), Sector 39-A, Chandigarh 160036, India; 3Central Building Research Institute, Council of Scientific and Industrial Research (CSIR), Roorkee 247667, India

**Keywords:** SARS-CoV-2, UVC, disinfection, COVID-19, aerosols

## Abstract

The World Health Organization (WHO) declared in May 2021 that SARS-CoV-2 is transmitted not only by close contact with infectious respiratory fluids from infected people or contaminated materials but also indirectly through air. Airborne transmission has serious implications for the control measures we can deploy, given the emergence of more transmissible variants. This emphasizes the need to deploy a mechanism to reduce the viral load in the air, especially in closed and crowded places such as hospitals, public transport buses, etc. In this study, we explored ultraviolet C (UVC) radiation for its ability to inactivate the SARS-CoV-2 particles present in aerosols and designed an air disinfection system to eliminate infectious viruses. We studied the virus inactivation kinetics to identify the UVC dosage required to achieve maximum virus inactivation. Based on the experimental data, UVC-based devices were designed for the sanitization of air through HVAC systems in closed spaces. Further, a risk assessment model to estimate the risk reduction was applied which showed that the use of UVC radiation could result in the reduction of the risk of infection in occupied spaces by up to 90%.

## 1. Introduction

The Severe Acute Respiratory Syndrome Coronavirus-2 (SARS-CoV-2) is responsible for the current pandemic causing millions of infections worldwide and over 2.6 million deaths due to COVID-19 [1]. Apart from taking such an enormous toll on human life, the pandemic has imposed unprecedented economic, societal, and healthcare burdens. While mass vaccination drives across the globe are helping to bring back normalcy, there is an urgent need to prevent virus transmission through aerosols to curb the airborne transmission and to allow the opening of public places and transport, such as schools, cinema halls, buses, trains, etc. [2]. There have been instances where reopening schools and offices, in the descending phase of the pandemic and without adequate measures of transmission-prevention, have backfired and led to outbreaks [3].

Many methods are being explored for the effective disinfection of materials and common-touch surfaces, such as alcohol-based disinfectants, copper surfaces, and soaps. However, there is no effective and safe method that can be deployed in closed workplaces to inactivate the SARS-CoV-2 virus in the air and intense research is being carried out in that direction [4,5,6,7,8,9]. The face masks are particularly effective in restricting transmission through aerosols and prevent the spread [10]. However, the common breach in face-mask practices is causing transmission of the virus within small groups and bigger gatherings. This suggests that control measures must be taken beyond personal hygiene. UVC radiation is already well known to effectively kill bacteria and viruses [11] and recent studies have shown that this method is effective in inactivating the SARS-CoV-2 virus [12,13,14,15,16]. UVC light causes the formation of pyrimidine dimers in the genetic material and inhibits transcription and replication [17,18]. Therefore, targeted damage of the essential components of the microbes could be an excellent strategy to attenuate the infectivity of viruses, especially the ones which transmit rapidly, such as SARS-CoV-2.

The WHO declared in May 2021 that SARS-CoV-2 transmission is airborne. Indeed, SARS-CoV-2 was detected in our studies in air samples near hospitals even earlier, and similar findings were also made by others [19,20,21]. Air-born transmission has serious implications for the control measures we can put in place and emphasizes the need for a reduction of viral load in the air. In this study, we explored UVC radiation for its ability to inactivate the SARS-CoV-2 particles present in the aerosols and design air purification systems based on the experimental data. The germicidal activity is heavily dependent on the UVC wavelength and the optimal dosage must be estimated for maximal antiviral activity. The 254 nm wavelength is conventionally used for anti-pathogen activity and causes maximal DNA damage (35104367). Therefore, a large amount of air can be passed through the ducts and exposure of the air for a short period of time should be sufficient to inactivate viruses. We studied the inactivation kinetics by using different dosages of UVC to identify the dosage required to achieve maximum virus inactivation. Devices using UVC light for the sanitization of air within the air-ducts of HVAC systems and circulating units were designed, developed and validated to deliver the relevant viricidal doses. Further, a risk assessment model to estimate the risk reduction was developed and analyzed which estimated that the use of a UVC-based air disinfection system could result in the reduction of the risk of infection in the occupied spaces by up to 90%.

## 2. Results

### 2.1. Design of Ultraviolet C (UVC) Disinfection System

Based on the market requirements, two types of systems were used—(i) an in-duct UVC-disinfection system and (ii) a stand-alone air-circulating UVC-disinfection system. These designs are described below.

*In-duct UVC disinfection system:* In-duct UVC systems work for the purpose of inactivating microbes in the airstream in a building or zonal ventilation system. The UVC lamp is a key component and the design and optimization of in-duct UVC systems revolves around various features of it. The in-duct system also depends on the output characterization, the required UVC dose for microbicidal activity, and energy consumption evaluations of UVC systems. The in-duct system we designed consisted of UVC lamps, fixtures, and ballasts. In the chamber, the access to the air by UVC radiation is uniform and so the UVC lamps could be fixed at any location including the air handling unit (AHU) (Figure 1A). While designing the system, we placed the lamp fixtures and ballasts either internally or external to the ductwork. The drop in pressure with external fitting was relatively lower than with internal fitting. In any case, the drop in pressure associated with UVC radiation was only marginal when the velocity of air was within the normal limits of 2–3 m/s (400–600 fpm). By applying these parameters, the modular systems were made and installed in the ductwork. The UVC radiation may not disinfect the air in the duct in one encounter but the recirculation of the air from the room will increase such encounters and give multiple UVC doses to microorganisms for maximum disinfection. Therefore, the re-circulation systems are more efficient than single-pass systems in which air is not recirculated. The characteristics of an air stream that can impact the design are relative humidity (RH), temperature, and air velocity. These factors may have an impact on microbial susceptibility in general; although membrane viruses such as coronavirus are more susceptible to higher temperatures because of the rapid drying of aerosols in hotter environments [22,23,24,25]. The RH factor is compensated as the UVC systems are installed after the air is conditioned and before delivery to the user.

In addition to this, we also considered the following parameters: (i) duct dimensions, (ii) duct materials, (iii) air temperature, (iv) amount of recirculated air, (v) amount of fresh air in recirculation, and (vi) flow rate. The different parameters of the proposed systems were X: length of the source in the *X*-axis, Y: height of the source surface in the *Y*-axis, Z: target surface in the *Z*-axis, and h: total installation height. The UVC intensity was evaluated against these parameters with respect to the flow rate. The required parameters such as flow rate, exposure time, amount of air recirculated in each cycle, installation space, and UVC intensity were optimized to obtain the required dosages for inactivation of the virus and maintain the constant fluence rate in the duct. The design of the in-duct UVC system (unidirectional and bi-directional) is shown in Figure 1B and Appendix A. It shows an enclosed space with a source to target the illumination surface in the z direction and the system was used in the closed indoor space. The principal design objective for an in-duct UVC air disinfection system was to create UVC energy distribution uniformly throughout a specified length of the duct or air-handling unit (AHU) to deliver the appropriate UVC dose to bacteria/virus/aerosol particles in the air moving through the irradiated zone with minimum system power, as shown in Figure 1C. Enhancing the overall reflectivity of the air inside the air handler or air duct improved the UVC-system’s performance by reflecting UVC energy back into the irradiated zone, thus increasing the effective UVC dose and maintaining the constant fluency rate. The flow rate was optimized and the change in air flow rate due to the obstruction caused by the UVC source in the flow was evaluated, and found to be negligible in the design. The designed and fabricated duct is shown in Figure 1D.

*Stand-alone recirculation air disinfection:* The recirculation UVC disinfection systems were installed in the smaller workplaces; each consisted of UVC lamps and fixtures in a housing containing a blower. The airflow in the recirculation units was in the range of 1.4–14 m^3^/min (50–500 cfm) and was suitable for small rooms or apartments only. Many recirculation units were portable and could be positioned on the floor or table, or mounted on wall or ceiling. Room recirculation units and upper air systems were installed to augment the in-duct systems or where in-duct installation was not feasible. The prototype stand-alone unit was tested under laboratory conditions and, based on the results, these units are currently installed for application in real-world scenarios. Due to the requirement of a compact size, the internal volume of the recirculation units does not usually allow extended exposure times to deliver the viricidal dosages during the transit of air through them. Hence, the units were carefully designed to create enough serpentine paths through the circulating devices within the volume irradiated by UVC light so as to deliver the viricidal dosages while the units remained compact and portable. The air that goes into the unit comes out sanitized to the extent of over 99% of virus-load reduction.

Both the in-duct and the stand-alone systems were designed for applications ranging from very small volume and low flow rate to high volumes and high flow rates for different applications such as lifts, toilets, classrooms, workspaces, offices, meeting halls and auditorium, etc. Depending on the application of the device (HVAC building vs. HVAC bus vs. a room) the configuration of the device varied but the dose of UVC per unit of air remained in the same range. The details of the different systems are as shown in Table 1.

### 2.2. Design of a Stand-Alone System to Measure the Effect of UVC on SAS-CoV-2 Survival in Air

A hermetically sealed chamber was designed to study the dosage of UVC required to inactivate the SARS-CoV-2 particles present in aerosols. This was the prototype system that was made for testing purposes against SARS-CoV-2. The volume of the aerosol generation and the volume of the UVC treatment are interrelated. The aerosols were created from virus suspension by using a nebulizer which produces mist by creating a low-pressure zone at the surface of the liquid which pulls up fine droplets from the liquid surface. The shape of the chamber was designed to be a rectangle with dimensions of 56 × 41 × 31 cm to ensure the uniform distribution of aerosols (Figure 1E,F and Appendix A). The inlet and outlet of the chamber were customized to fit to the nebulizer pipe and sample collection tube, respectively. The regulator was installed to control the UVC intensity. The air filtration unit was connected to the chamber to suction the air from the chamber and a filter was placed in the outlet pipe to collect the aerosols on the surface of the filter membrane.

### 2.3. UVC Air Disinfection System Efficiently Inactivates Aerosolized SARS-CoV-2

A number of different methods are used to entrap the viruses. These include impactors, electrostatic precipitators, filters, etc. [26]. The transmission of COVID-19 has been primarily estimated through collecting specified volumes of air on filters followed by estimation of the viral load through quantitative real-time polymerase chain reaction (qRT-PCR) [27,28]. However, such methods only look at the presence of virus particles in the air samples and do not provide any information on infectious dose. Since our primary goal was to evaluate the effect of UVC radiation on the infectivity of airborne SARS-CoV-2, we used the prototype system described above to entrap airborne SARS-CoV-2 and assess the effect of UVC irradiation (design of device and experimental set-up is shown in Figure 2A and actual device is shown in Figure 2B). For each cycle of aerosol generation, a virus suspension containing 1.5 × 10^7^ pfu was nebulized into the chamber. Two minutes were sufficient to produce aerosols from approximately 90% of the loaded virus stock in the nebulizer. The nebulized SARS-CoV-2 was either collected directly (control) or exposed to UVC light and then collected by trapping it onto a gelatin filter using an air sampling device that channeled the air through the gelatin filter (Figure 2B). However, the virus particles recovered from gelatin membrane were not infectious as assessed on the highly receptive cells (Vero-E6-TMPRSS2). Here, we deliberately used the samples for the infection of Vero cells and allowed them to amplify the virus before detection by qRT-PCR method. The direct detection of infectious particles by plaque forming unit (pfu) assay could underestimate the infectious virus particles. To test whether gelatin had any intrinsic virus inhibitory properties, the virus suspension (10^6^ pfu/mL stock) was directly added to the gelatin filters and incubated for 10 min. The viruses present on the membrane were infectious and viral genomic RNA was detected in the culture supernatants after 24 h of incubation. (Figure 2C). Collectively, these findings indicate that SARS-CoV-2 viruses are probably quite fragile and susceptible to air-drying and collision with solid surfaces.

### 2.4. Testing the Efficiency of Stand-Alone UVC Air Disinfection System against SARS-CoV-2 Virus

As mentioned above, the traditional device such as an air sampler was not suitable for the analysis of infectivity of SARS-CoV-2 due to the mechanical disruption of the virus particles by collision with the filter. Therefore, we aimed to develop a tool whereby we could expose the SARS-CoV-2 to UVC and then collect the virus to analyze its infectivity (Figure 3A). To circumvent the issues related to the trapping of viral particles through an air filtration device, we took inspiration from the natural mode of SARS-CoV-2 transmission. The droplet nuclei containing SARS-CoV-2 are usually inhaled by healthy individuals and deposited inside the airway tissue. We utilized the aerosol’s tendency to settle on surfaces such as glass and steel to trap the viruses [29]. SARS-CoV-2 was nebulized, and then the nebulized particles were collected in the quartz tube in an air-tight aerosol chamber connected to the nebulizer (Figure 3B). The open quartz tubes were placed inside the chamber to collect the aerosol sample. The device was equipped with a 30 mW UVC light tube enabling the exposure of the trapped viruses in the center of the device. After UVC exposure, the quartz tube was rinsed with cell culture media to recover the viruses. The aerosols were exposed to UVC at different doses ranging from 2.8 mJ/cm^2^ to 16.8 mJ/cm^2^. A UVC dose of 0.28 mJ/cm^2^ was sufficient to inactivate 99.2% of SARS-CoV-2 whereas 0.56 mJ/cm^2^ resulted in 99.8% inactivation of SAR-CoV-2 (Figure 3C). Thus, we concluded that 0.28 mJ/cm^2^ was sufficient to meaningfully inactivate the aerosolized virus. The disinfection of the virus was most likely due to chemical modification of the genomic RNA but we also investigated whether UVC had any effects on the spike protein of SARS-CoV-2. To this end, we used SARS-CoV-2 spike protein ectodomain (spike-6P) stabilized in a pre-fusion state [30]. The UVC irradiation equivalent to 0.28 mJ/cm^2^ enhanced the binding of serum antibodies but not the S309 antibody which recognized the conserved epitope in RBD [31] (Figure 3D). The UVC treatment did not break the peptide chain and only one intact band appeared in SDS-PAGE (Figure 3E). This data suggested that UVC radiation alters the conformation of the spike protein and may thus affect the entry of the virus. In summary, the above-described observations suggest that UVC radiation can efficiently inactivate airborne SARS-CoV-2 and could be used for the disinfection of air.

### 2.5. Risk Analysis of Infection and Reduction of Risk

The UVC-based air disinfection system resulted in a rapid loss of infectious SARS-CoV-2 particles and, based on this data, we next assessed the reduction of transmission risk when this system is used in a real-world scenario. The transmission of pathogenic microorganisms is a complicated process. It comprises pathogen features, the number of particles produced in a potentially pathogenic host, how effectively the pathogen survives or remains viable outside that host, and the immune system of a person who is exposed to the pathogen. For decades, the Wells–Riley model has been primarily employed for this purpose [32].
$$P=1-e^{-n\;}$$
where *P* is the probability of infection for susceptible persons and n is the number of quanta inhaled. The quanta inhaled is influenced by the average quanta concentration (*C*_avg_, quanta/m^3^), volumetric breathing rate of an occupant (Q_b_, m^3^/h), and the duration of the occupancy (D,h).
$$n=C_{\mathrm{avg}}Q_{b}D$$

The airborne quanta concentration rises with time from zero to one minus exponential. The time-dependent airborne concentration of infection quanta depends on loss rate coefficients which are the sum of ventilation rate, deposition on to surfaces, virus decay and filtration or air disinfection efficiency [33].

To quantify the impact of the virus concentrations, we considered an example of a secondary school classroom which consists of 40 students and a faculty member. Let us consider a scenario where the faculty member is infected with SARS-CoV-2, assuming no susceptible student is wearing a mask and they have not been vaccinated. The breathing rate of the faculty member as per talking activity is 1.1 [33] and the quanta emission rates for this activity are 9.7 [34]. The recommended rate of air circulation for a school building is five air changes per hour [35]. The surface deposition loss rate is considered as 0.3 (1/h) [36,37]. Fears et al. [38] observed no virus decay in virus-containing aerosol for 16 h at 53% relative humidity, but van Doremalen et al. [29] calculated the half-life of airborne SARS-CoV-2 to be 1.1 h, corresponding to a decay rate k = ln(2)/t_1/2_ of 0.63 1/h.

As per the investigation, the efficiency of the UVC dosage of 0.28 mJ/cm^2^ led to a 99.2% inactivation while a UVC dosage exposure of 0.56 mJ/cm^2^ resulted in 99.8% inactivation of SAR-CoV-2. Considering the UVC disinfection system efficiency to be 99.5% as an average of the two, the risk of infection and reduction in risk calculation for a classroom with 6 h of operation is presented in Figure 4. Without UVC disinfection, after 6 h of occupancy in the classroom, 13 students out of 40 could become infected with SARS-CoV-2 (Figure 4). In comparison, after implementing a UVC disinfection solution, the risk of infection is reduced by 90%.

## 3. Materials and Methods

### 3.1. Design of UVC Disinfection System

Both the systems were developed using a commercial bill of material (BOM). The certified UVC sources, aluminum plates with surface treatment, and commercial grade fasteners and fittings were used. The UVC lamps used in the system were of 254 nm wavelength and were obtained from a commercial source. The UVC was measured using a Lutron UVC 254 light meter. The qualified power supplies, Teflon wiring with extra sleeves, were used as a safety measure. The developed system also met all the commercial safety, environmental and electromagnetic induction (EMI) directives. The systems were robust in design and had satisfactory performance to inactivate coronavirus at the installed places. All the parameters considered for designing the system are listed in Table 1. The 3D model in solid works was developed followed by the flow analysis. The intensity of UVC was optimized using TracePro software. The ray analysis was carried out considering the radiometry. Both the in-duct UVC air disinfection system and the stand-alone air disinfection system were designed according to the required optimum dosages, flow rates and duct dimensions. The system was designed in such a way that it did not affect the flow rates through the delivery duct and the UVC systems were mounted along the side wall in the in-duct UVC air disinfection system. The in-duct system comes in two configurations, i.e., an extraction- and retraction-type mechanism for the overhead delivery ducts and the fixed type (refer Figure 1B,D) for the separated delivery ducts. The stand-alone system (Figure 1F and Appendix A) is for use in public places frequented by human life and has a fixed UVC source and is designed to meet the standard COVID-19 ventilation guidelines and 10 ACH (air changes per hour). Both the systems were adequately sealed and UVC leakage protection was ensured through the optimum sealing. In order to increase the effectiveness of the UVC source, aluminum polished plates were used and the irradiated zone had high finish aluminum surface for efficient delivery.

### 3.2. SARS-CoV-2 Virus Preparation

All the experiments involving the handling of the SARS-CoV-2 virus were performed in the BSL3 facility according to the institutional biosafety guidelines and institutional ethics guidelines. The SARS-CoV-2 strain used in the study was isolated from a patient and cultured using the VeroE6 cell line as per the established methods [39,40]. The SARS-CoV-2 (Wuhan strain) was confirmed by whole-genome sequencing and the sequence was submitted to the gene bank (EPI_ISL_11450498). An aliquot of the virus from passage-1 was used to inoculate the 25 mm cell-culture flask containing Vero E6 cells with 80–90% confluent cells in 5 mL medium. The virus growth was monitored regularly. After extensive cytopathic effect, virus suspension was harvested, clarified and aliquoted in microcentrifuge tubes for long storage at −80 °C until further use. The viral load was estimated by quantitative real-time PCR by using a 10 µL sample diluted in 100 µL growth medium. Same virus stock was used for all the experiments.

### 3.3. Aerosol Generation, Sample Collection and Infection of Cells by Aerosol Samples

Before setting up the experiments, the device was cleaned and disinfected using 70% ethanol. The working 0.5 mL virus stock was freshly thawed for aerosolization. The sterile quartz tube was placed inside the UVC chamber and the chamber was closed from all sides. A measure of 0.5 mL of virus solution was loaded into the sample holding cup of the nebulizer and the mouthpiece was then attached to the inlet tube of the UVC device. The aerosols were created and allowed to accumulate in the device chamber for 2 min. UVC irradiation was carried out for the required period of time by using a UVC light switch outside the bio-safety cabinet. After UVC treatment, the quartz tube was removed carefully from the UVC chamber, and immediately 0.5 mL growth media was added to rinse the surface of the tube. A 50 µL sample was then used to infect the cells in a 48-well tissue culture plate containing 5 × 10^4^ Vero-E6-TMPRSS2 (JCRB1819) cells per well [41]. The infection was carried out for 1 h with intermittent swirling. After incubation, cells were washed with 200 µL sterile PBS, and a fresh 200 µL growth medium was added to the wells, and the plate was further incubated for 24 h at 37 °C in the atmosphere of 5% CO_2_.

### 3.4. Quantitative Real-Time PCR (qRT-PCR)

The Vero cell supernatants were harvested from the test plate after the required incubation. RNA was isolated from 140 μL supernatant for qRT-PCR-based analysis. The RNA isolation by Qiagen RNA isolation kit and qRT-PCR by GCC Biotech was performed using the manufacturer’s instructions. The qRT-PCR was performed using Bio-Rad system CFX96 Real-Time System. The extent of virus inactivation was calculated by quantifying the viral RNA in the respective virus cultures from UV-treated aerosols and non-treated aerosols.

### 3.5. SARS-CoV-2 Spike Protein Expression

The spike protein preparation was carried out as described in an earlier study [42]. Briefly, expi293 cells were cultured in 30mL Erlenmeyer flasks in freestyle expression media in a humidified incubator with 5% CO_2_ at 37 °C temperature. The shaking speed was 130 rpm/min. The cells were seeded 1 day before transfection to achieve 3 × 10^6^ cells/mL at the time of transfection. The cells were transfected using polyethyleneimine (PEI) transfection reagent (Polysciences, Warrington, PA, USA). Transfected cells were further incubated at constant shaking speed for 4 days before harvesting the culture supernatant. The supernatant was filtered through 0.45 u filters and passed slowly through HisPur Ni-NTA resin column (Thermofisher Scientific, Waltham, MA, USA) with the speed of ~0.1 mL/min. The nickel beads were washed twice with washing buffer (NaCl) followed by elution with imidazole (150 mM). The eluate was then dialyzed overnight in PBS and protein was concentrated using an Amicon Pro centrifugal filter (Millipore, Burlington, MA, USA). The protein was quantified using BCA Protein Assay Kit (Thermofisher Scientific).

### 3.6. Antibody (S309) Expression and Purification

The heavy and light chain-expressing constructs were a generous gift from Dr. Raghavan Varadarajan. The expi293 cells were co-transfected by light and heavy chain-expressing plasmids by using PEI. The culture supernatant was harvested after 5 days post-transfection. The clarified and filtered supernatant then was passed through Protein A column (Sigma Inc, Marlborough, MA, USA) to capture IgG onto the beads. The eluted IgG was then dialyzed in PBS and concentrated to 1 mg/mL.

### 3.7. SDS-PAGE

A sample of 2 μg protein was loaded onto 10% SDS gel followed by staining with Coomassie brilliant blue dye.

### 3.8. Enzyme-Linked Immunosorbent Assay (ELISA)

A sample of 1 μg/mL spike protein in PBS was coated overnight in 96-well plates, followed by washing and blocking the plate with PBS containing 5% FBS plus 1% skimmed milk for 1 h. After blocking, S309 or vaccine serum was added in serial dilution and incubated for 2 h at room temperature. The antibodies were washed thrice using PBS; then, anti-human secondary antibody diluted at 1:3000 was added for 1 h. The plate was washed four times with PBS containing 0.1% tween 20 followed by addition of TMB substrate. The reaction was stopped by adding 0.3M H_2_SO_4_. The optical density was recorded by reading the plate at 450 nm.

## 4. Discussion

In this study, we explored the effectiveness of UVC light to inactivate the SARS-CoV-2 virion particles present in aerosols. UVC radiation has long been known as a sterilizing agent for microbes in laboratories, hospitals, and the food industry, and is regularly used to sterilize various materials. Unlike bacterial, parasitic, or fungal pathogens, viruses have little genetic material with a simpler physical structure and thus are more susceptible to UV-induced destruction [11]. UVC is considered to be the most effective band of the UV spectrum and efficiently catalyzes the formation of photoproducts in DNA [43]. Thymidine dimers irreversibly interrupt the essential processes of replication, transcription, and translation culminating in the inability of virus particle to establish infection. This study confirms that UVC rapidly inactivates the SARS-CoV-2 virions present in the air and attempts to utilize its power to mitigate the airborne transmission of viruses.

SARS-CoV-2 is a highly infectious virus which is evident from the rapid spread of this virus from close or continuous contact with the infected person or from common touch surfaces. Our study using the air sampler, however, revealed that the virus in the aerosols is quite sensitive to the physical collision of the aerosols onto a solid surface and desiccation. Both these factors could affect the membrane integrity or spike protein structure and likely account for the virus inactivation. The results suggest that strong air currents might help with virus inactivation in the air and UVC would further reduce the viral load to a non-infectious baseline.

In 2020, when the world witnessed the huge numbers of COVID-19 infections and the death toll, there were no vaccines or effective anti-viral drugs available. With the arrival of vaccines in 2021, we are poised to have better control over the pandemic and are in a far better condition to prevent the new infections or disease severity. However, new studies reveal that the vaccine-mediated immunity wanes over time and increases the chance of re-infection [44,45]. Moreover, the elderly and immunocompromised people are more susceptible to re-infections even after vaccination. Getting back to normal life is important for sustainability and will require effective methods to inactivate the SARS-CoV-2 virions from our surroundings to prevent the chances of infection. As SARS-CoV-2 is airborne and highly infectious, rapid and effective air sterilization could substantially contribute to reducing the viral load in the air. Our study reveals that UVC radiation effectively inactivates SARS-CoV-2 in highly concentrated virus-loaded aerosols in the air. Exposure to UVC radiation equivalent to 0.28 mJ/cm^2^ is enough to reduce >99% of viruses and therefore could be used to devise strategies of air-purification. One of the limitations of this system is that UVC lamps do contain some mercury and the relevant guidelines mandate the adherence of national and international policy for its disposal [46,47].

In conclusion, air-disinfection technology based on UVC is a promising alternative to implement disinfection protocols in crowded places. Workplaces such as large offices, hospitals, etc., are potential settings that are vulnerable to rapid transmission. UVC systems are generally used in combination with HEPA filters which surely enhance the pathogen-filtering capacity but also can increase the cost and maintenance of the systems. The viral pathogens, especially respiratory viruses, could be inactivated by simple UVGI systems. The facile design of our systems presented in this study and the methodologies are accurate for the sizing of UVC systems.

## Figures and Tables

**Figure 1 pathogens-12-00419-f001:**
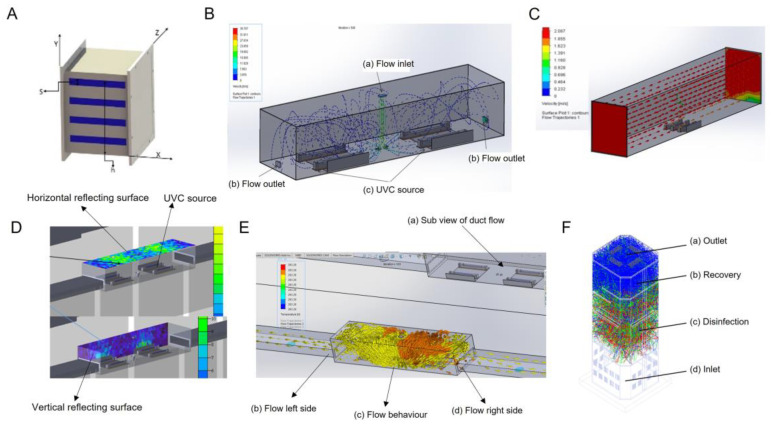
Configuration of UVC disinfection systems. All systems were considered rectangular. A mathematical equation and geometrical aspects were considered and simply derived from UVC lighting reference. Based on that, the systems were designed and validated for the desired intensity of the UVC light in the designated space. UVC was measured using Lutron UVC 254 light meter. (**A**) In-duct UVC air disinfection system in a cartesian coordinate system with flow in x direction, Illumination in z-direction and source (S) installation in y-direction. The length is denoted in millimeters (mm). (**B**) The large size UVC duct with bi-directional flow arrangement with flow inlet (a) at top for entry of infected/recirculated air, (b) flow outlets for exit of disinfected air, (c) UV-C source fixtures. The blue dotted lines show the flow trajectory of air. (**C**) Unidirectional flow system with the UVC source at bottom and air being treated by disinfection volume. The scale shows the velocity in m/s. (**D**) The irradiance plot and flow behavior in the duct with horizontal target surface (a), vertical surface (b), and UVC source (c). (**E**) bidirectional flow behavior of air from the center of duct. (**F**) Stand-alone air disinfection system with flow in vertical direction. Shown is the inlet for air (a), UVC treatment compartment (b), the recovery area (c), and outlet for the disinfected air (d).

**Figure 2 pathogens-12-00419-f002:**
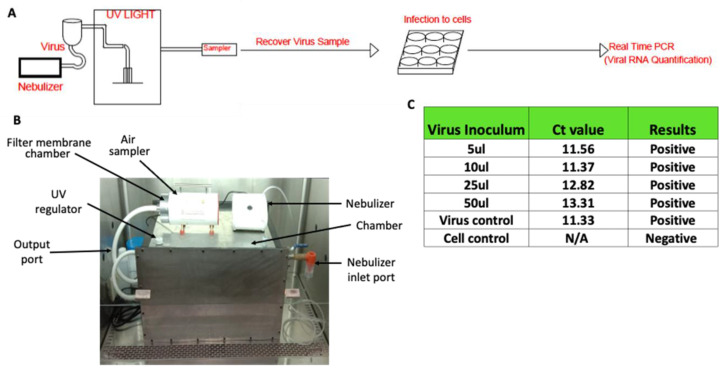
Workflow of SARS-CoV-2 aerosol generation and detection of virus in air. (**A**) The schematic of the procedure of aerosol generation, virus sampling, infection and detection of virus growth. (**B**) The UVC treatment and air sampling device. (**C**) The effect of gelatin on the viability of SARS-CoV-2. The virus supernatant was applied onto gelatin membrane and recovered by dissolving membrane in PBS to assess infectivity in Vero-TMPRSS2 cells. The Ct values for original virus supernatant (virus control) and gelatin-recovered virus are indicated in the table.

**Figure 3 pathogens-12-00419-f003:**
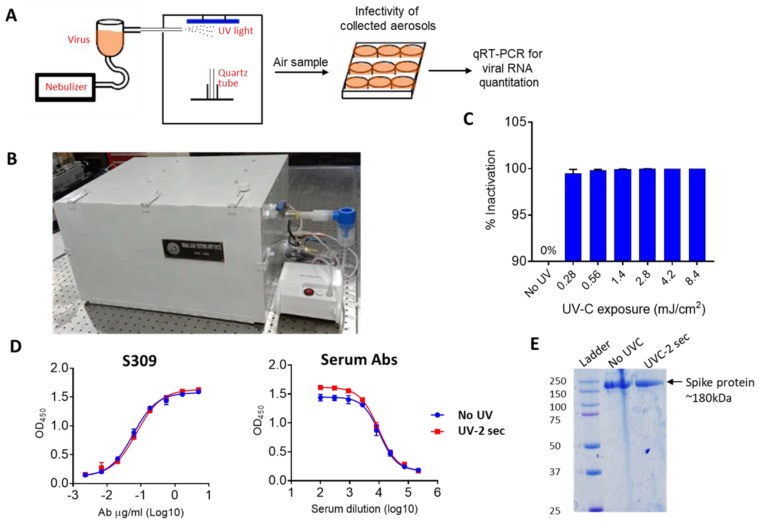
Inactivation of SARS-CoV-2 present in air. (**A**) The schematic of aerosol generation virus sampling, infection and detection of virus growth. (**B**) The stand-alone UVC-based device without the air sampling connector to sample air containing aerosolized virus inside the chamber. (**C**) Inactivation of SARS-CoV-2 by UVC treatment. The recovered virus aerosols from the quartz tube after UVC treatment were resuspended in growth medium and used for infection of Vero-TMPRSS2 cells for 48 h. Cell supernatants were analyzed for viral RNA. Percent inactivation of virus at various UVC exposures with reference to no UVC treatment was recorded. (**D**) The binding of S309 or serum antibodies to spike protein is shown in dose-dependent manner. (**E**) The spike protein with or without UVC treatment in SDS-PAGE.

**Figure 4 pathogens-12-00419-f004:**
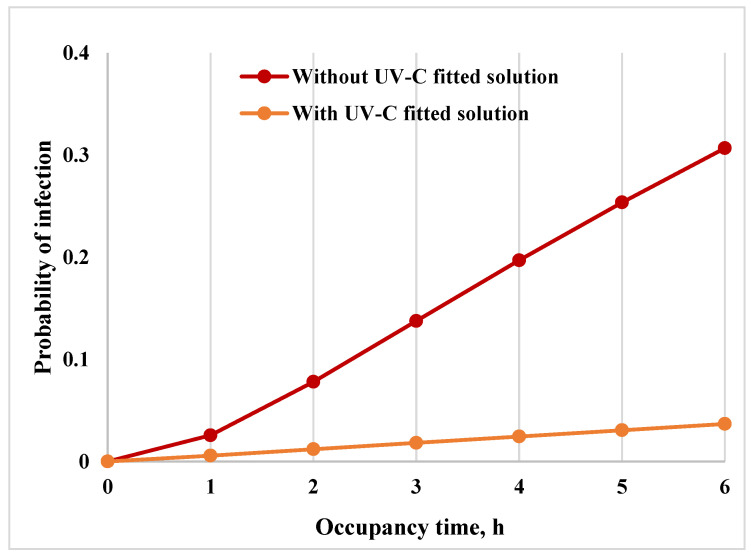
The mathematical model of the risk reduction by UVC treatment of air. Based on the results of SARS-CoV-2 virus inactivation using a stand-alone device, the risk reduction in the classroom was modelled. Shown is the % reduction of risk with the function of exposure time.

**Table 1 pathogens-12-00419-t001:** Types of UVC disinfection systems.

S. No	Parameters	In-Duct UVC Disinfection Systems for HVAC Buildings	In-Duct UVC Disinfection Systems for HVAC Buses	Stand-alone UVC Disinfection System
i.	Flow rate (CFM)	>30,000	<3000	<300
ii.	Applicable space(ft^2^)	>1000	<350 (customizable)	<150
iii.	Shape and size	Rectangular >300 mm	Rectangular <300 mm	Rectangular <1500 mm
iv.	Wattage (W)	>100	<100	<70
v.	Weight	<2.5 Kg	<1.2 Kg	<3.0 Kg
vi.	UVC source (nm)	254	254	254
vii.	Length of the delivery duct (m)	>10	<2	<0.5
viii.	Exposure time (s)	1.1	1.14	1.1
ix.	Dosage	>1.6 mJ/cm^2^	>1.3 mJ/cm^2^	>1.3 mJ/cm^2^

## Data Availability

The data are not publicly available as patents based on the data presented in this paper have been filed.

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
