# Peer review of "UVC-Based Air Disinfection Systems for Rapid Inactivation of SARS-CoV-2 Present in the Air"

_pathogens, 2023, doi:10.3390/pathogens12030419_

Round 1
Reviewer 1 Report
It is very important and useful to develop rapid disinfection systems. It will reduce the infection risk of occupants in indoor environments.
The manuscript reported testing of two UVC devices "within air-ducts" and "circulating unit for use within occupied spaces." The organization of the manuscript is not good, and making it hard to read. My comments are below:
1. Fig 1 shows the configurations of the two devices. The original purpose should give the reader what devices were tested. Unfortunately all these six figures do not achieve that purpose. Fig A, I can't see the dimension (supposed to be there). If the devices were made, why not show a few photos. If the authors want to show the CAD drawing, it is fine, BUT remove those simulation pattern. For instance Fig F, I hardly can see anything. Besides, make all the figures larger, if there is no page limit. Fig D, I can't see anything at all.
2. Ln 111-112. How to measure the UV irradiation? What instrument used? How many directions were measured?
3. Table 1. The caption "Types of UV-C disinfection systems." It showed many parameters such as Flow rate > 30,000 CFM for In-duct UV-C. I am a bit confused. Are these the experimental testing parameters? If yes. Where are the disinfection results. The authors designed the In-duct UVC system (according to Fig 1), but I can't see any disinfection results?
4. Ln 204-210. Needed to be written. This paragraph should focus on the testing. Suddenly it mentioned UVC has several advantages over existing chemical methods. This should be mentioned in the Introduction. In fact Ln299 to 311 have the similar problem.
5. Ln 162-193. The last sentence "Collectively, these findings indicate that SARS-CoV-2 viruses are probably quite fragile and susceptible to air-drying and collision onto the solid surfaces. If the experiments were failed, I don't think it needed 30 lines to explain.
6. Ln 211-220. This described an improved set-up. However, Fig 2 A is too small. Please indicate teh quartz tube, 30mW UV light tube in the Figure. The light tube enabling exposure of the trapped viruses. Show it in the Fig.
Author Response
It is very important and useful to develop rapid disinfection systems. It will reduce the infection risk of occupants in indoor environments.
The manuscript reported testing of two UVC devices "within air-ducts" and "circulating unit for use within occupied spaces." The organization of the manuscript is not good, and making it hard to read. My comments are below:
- Fig 1 shows the configurations of the two devices. The original purpose should give the reader what devices were tested. Unfortunately all these six figures do not achieve that purpose. Fig A, I can't see the dimension (supposed to be there). If the devices were made, why not show a few photos. If the authors want to show the CAD drawing, it is fine, BUT remove those simulation pattern. For instance Fig F, I hardly can see anything. Besides, make all the figures larger, if there is no page limit. Fig D, I can't see anything at all.
Response: We agree with reviewer about the image quality and sizes. We have improved the figures to our level best considering the space constraints. We have also added the actual photos of the devices made.
- Ln 111-112. How to measure the UV irradiation? What instrument used? How many directions were measured?
Response: UV was measured using Lutron UVC 254 light meter. It was measured in 3-directions. We have added these details in the revised version of the paper (In Methods section).
- Table 1. The caption "Types of UV-C disinfection systems." It showed many parameters such as Flow rate > 30,000 CFM for In-duct UV-C. I am a bit confused. Are these the experimental testing parameters? If yes. Where are the disinfection results. The authors designed the In-duct UVC system (according to Fig 1), but I can't see any disinfection results?
Response: The residence time and the intensity of UV are important parameters and determine the dosage of UVC per litre of air passed through the system. The designs of the systems vary depending on the application but importantly maintaining the required dosage of UV is key factor given different flow rate of air and shape of the device. What is most important in this is the calculation of UVC dose required to inactivate >99% virus in the air. We have used one prototype device to calculate that in the laboratory conditions which showed that the dose of 1.2mJ/cm2 required for that purpose (results shown in figure 3C). Now, depending on the application of the device (HVAC building vs HVAC bus vs a room) the device configuration changes but the dose of UVC per unit of air remains the same.
We understand the concern of the reviewer and we have added these details in the revised version for better readability.
- Ln 204-210. Needed to be written. This paragraph should focus on the testing. Suddenly it mentioned UVC has several advantages over existing chemical methods. This should be mentioned in the Introduction. In fact Ln299 to 311 have the similar problem.
Response: We have now corrected the language deleting unnecessary sentences.
- Ln 162-193. The last sentence "Collectively, these findings indicate that SARS-CoV-2 viruses are probably quite fragile and susceptible to air-drying and collision onto the solid surfaces. If the experiments were failed, I don't think it needed 30 lines to explain.
Response: We have re-written this para to make it succinct. However, existing description is necessary to emphasize that the method of virus collection from membrane is not appropriate and can miscalculate the effect of UV on virus inactivation.
- Ln 211-220. This described an improved set-up. However, Fig 2 A is too small. Please indicate teh quartz tube, 30mW UV light tube in the Figure. The light tube enabling exposure of the trapped viruses. Show it in the Fig.
Response: We agree with reviewer and we have now improved the figure quality in the revised version.
Reviewer 2 Report
1) Manuscript lacks novelty, this particular topic has been well studied in the literature.
2) UVC is well known to have the germicidal properties from almost 100 years. Can author explain why they choose UV source which emit at 254 nm? What is its importance in terms of wavelength? Please include it in the paper.
3) Mercury lamps are already in market for almost 50 years which means their is a reason why they are not used for disinfection in public places. Can authors comment on that?
4) I believe authors are aware of Minamata Convention? What is the relevance of this work in that context?
5) Also, authors should talk about all the safety aspects related to Mercury source in the paper.
6) Authors should not claim anything which they have not studied. Line 410-411 is unwarranted and does not any value to the paper.
7) I will recommend authors to put actual pictures of the different configuration of UV-C disinfection systems described in this paper.
8) I believe they have done simulations too. Please explain them in detail.
Author Response
Reviewer #2
- Manuscript lacks novelty, this particular topic has been well studied in the literature.
Response: We thank reviewer for pointing it out and we also share the same opinion. However, many of the papers have analysed the effect of UV radiation on germicidal/anti-viral activities at the in vitro level or the laboratory setting. In this study we are moving forward to implement the UV-based system for practical benefits in real world scenario. The systems have been installed at various places and we plan to monitor in time to what extent it will benefit the prevention of air borne infections.
2) UVC is well known to have the germicidal properties from almost 100 years. Can author explain why they choose UV source which emit at 254 nm? What is its importance in terms of wavelength? Please include it in the paper.
Response: We have re-written this paragraph to make it succinct and added the relevant references. 254nm wavelength UV is the most effective in modifying nucleic acid and thereby inactivating the pathogens and that why it was used in this study. We have explained it in the revised paper.
3) Mercury lamps are already in market for almost 50 years which means their is a reason why they are not used for disinfection in public places. Can authors comment on that?
Response: Mercury lamps are not as effective in anti-microbial properties. In our study we have used ultraviolet (254nm) which is highly effective in anti-microbial activities. We insist the proper installation to avoid leakage of UV radiation and proper disposal after the life of UV tubes to avoid mercury contamination in the environment and ecosystem. With these guidelines UV would be very effective in mitigating SARS-CoV-2 transmission via air.
4) I believe authors are aware of Minamata Convention? What is the relevance of this work in that context?
Response: We agree that in our system small amounts of mercury is present in the UV tubes. There are defined guidelines to dispose mercury as and when UV light s are non-functional. We recommend users to follow the strict guidelines mentioned in connection with mercury disposal. However, we feel that this topic is out of the scope of the paper and adding the details on it will impair the readability of the paper.
5) Also, authors should talk about all the safety aspects related to Mercury source in the paper.
Response: The mercury present is in very small amount in UVC tubes. We have added the description regarding its safety aspects and cautioned the reader about the disposal while providing references on that topic.
6) Authors should not claim anything which they have not studied. Line 410-411 is unwarranted and does not any value to the paper.
Response: We have removed the statement in the revised version; although it is a well-known fact.
7) I will recommend authors to put actual pictures of the different configuration of UV-C disinfection systems described in this paper.
Response: The actual photographs has been put in figure 1 A, B, C
8) I believe they have done simulations too. Please explain them in detail.
Response: We have described about it in the Methods section of revised manuscript.
Reviewer 3 Report
The authors present work on the efficiency of UV_C light to attenuate SARS-CoV-2 transmission, which is timely and relevant. However, the data presented here appears very disjointed and some of the conclusions are not as clearly supported by the results as needed.
Minor:
L20: What is a supposed to be crowded place?
L39: reopen the schools? Resume schooling?
L53: It can be argued that SARS-CoV-2 does not transmit rapidly, considering the relatively low R value. I suggest rewording this statement.
L92: Please add references. There is literature on the specifics of increased aerosol stability of SARS-CoV-2 in different humidity and temperature
L109: Grammar?
L201 onwards: all cursive letters?
What variant of SARS-@ was used? Accession number? How do the authors obtain pfu when they say the load was measured by PCR? Was there a standard curve?
Major:
1. There are English grammar mistakes throughout the manuscript, which should also be proof-read for consistency of verbiage. Example: Abstract L 17: “by touch and close contact, but also…”. Example: Air – borne, airborne, air borne. This is in itself only minor but adds up throughout and then becomes a hindrance to the reader.
2. L44: The way this is stated suggests that no one is working on UV-C irradiation systems. This is not true, as many commercial companies and other labs are utilizing such systems or developing them. The authors should present a more dedicated review of the relevant literature. Some examples may include:
https://www.ncbi.nlm.nih.gov/pmc/articles/PMC9014471/
https://www.ncbi.nlm.nih.gov/pmc/articles/PMC9538331/
https://www.ncbi.nlm.nih.gov/pmc/articles/PMC9411833/
https://www.ncbi.nlm.nih.gov/pmc/articles/PMC8605003/
https://www.ncbi.nlm.nih.gov/pmc/articles/PMC8522489/
https://www.ncbi.nlm.nih.gov/pmc/articles/PMC9437662/
This can be addressed be rewording the statements and presenting some of the new knowledge on what other groups have already established for UV-C.
3. Repeating the sane language in the abstract and introduction suggests sloppy writing
4. There are multiple conclusions drawn for the in-duct systems: Example: “reflection increases exposure”. To me, it is not clear if this is based on previous data, common knowledge, or actual results presented here in the manuscript. If the former, there are references missing. If the latter, the authors need to actually present data on this and not simply state it. The figure 1 does not help in understanding how the authors arrive at the conclusions. The figure is also difficult to read (low resolution?) and to understand. Perhaps the authors can improve the legend and explain if these are just schematic representations or actual mathematical models. Was math performed to arrive at the final designs? Any validation?
5. L136: Please explain if this means the authors did the testing. Can data be shown for this? Is this a commercial product which the public can buy already or do the authors mean they themselves are using it going forward?
6. L143: Can these results actually be shown? How was this measured, and is this a result first presented here?
7. L167: The authors need to also reference the literature in which actual infectious virus was isolated by plaque assay or TCID50 titration.
8. There is a methodical concern in how the authors measure infectious virus samples. Obtaining infectious virus from gelatin filters should not be a problem. It is highly unusual to measure virus by qRT after inoculating cells after 48h. If the authors would try to actually perform plaque assays, they may be more successful in recovering virus. At least they should describe the limit of detection of the method they use and provide more details how this method is used to arrive at pfu/do comparisons between values. The conclusion drawn in L203, therefore, may be wrong, and the inability to detect virus is a result of a too high assay detection limit. If the authors insist on PCR, they should at least look at sgRNA, instead of gRNA, which would more readily support virus replication. Showing gRNA, without showing the same analysis pre-incubation of the cells, could simply mean there is viral RNA present, but not infectious virus.
https://www.ncbi.nlm.nih.gov/pmc/articles/PMC7121658/#!po=25.0000 aerosol stability of SARS-CoV-2
https://www.ncbi.nlm.nih.gov/pmc/articles/PMC9413705/ gel filters able to capture virus from infected animals
9. L232: More experiments would be needed to draw this conclusion, or it has to be better explained why the authors assume so. The authors could actually test if entry is different, but no such data is shown. What this data means for the systems presented here is also unclear. Perhaps a move to supplement would be better.
10. Figure 2D and E: There are no statistics shown.
Round 2
Reviewer 1 Report
The authors still not fixed the issues for Fig 1. For instance, "A", how many mm is still not fixed.
Author Response
Response: We have now fixed figure 1 in the revised version.
Reviewer 2 Report
The author has worked on all the comments.
Author Response
Response: Thank you
Reviewer 3 Report
I thank the authors for their responses, and some of my issues have been resolved now. Some remain:
Minor:
L15: suggestion: “infected people” or “an infected person”
L19: suggest removing “supposed to be” as this suggests that these places may actually not be crowded.
L39: please fix grammar
L47: perhaps this is on purpose, but do they authors want to separate between UV and UVC at specific instances, or is it all referring to UVC?
L54/55: Are they the first to have shown so? Or is this just self-citation?
L68: Is quantize the correct word here?
L74: The authors may wish to keep UVC or UV-C consistent throughout
L385: Sentence?
Major:
1. The paper is still difficult to follow. I would suggest rearranging and starting with the experimental data which informed the design of the devices, before introducing the device.
2. The language is still at times incorrect and should require further revisions for readability and correctness.
3. L140 – 149: It may be prudent to move this section after the actual testing is shown. As it is, the authors keep referring to the data they present at later stages in the manuscript, which is very confusing for the reader to follow.
4. The authors should explain in the methods how their qRT method to measure virus actually translates to pfu/ml, which they use for the stocks.
5. L232: More experiments would be needed to draw this conclusion, or it has to be better explained why the authors assume so. The authors could actually test if entry is different, but no such data is shown. What this data means for the systems presented here is also unclear. Perhaps a move to supplement would be better.
Response: Yes, we agree with the reviewer that more experiments will be needed to flesh out the details. However, to simplify, we took recombinantly expressed pure spike ectodomain and given UV or no treatment. In both cases, band pattern was similar on SDS-PAGE in reducing conditions. If peptide chain is broken then we must get more than 1 bands in this condition. As we don’t see that happening, the inference is that peptide chain was intact post UV-exposure. This result can be extrapolated to the virus setting and we have described the implication of this in the manuscript. In conclusion, the UVC inactivates the virus not only by chemically modifying its genomic RNA but also damaging its Spike protein. The damaged spike would impair the virus entry.
Reviewer response: Thank you. The authors should provide additional data to strengthen their claim that the spike conformation is changed or provide better explanation in the manuscript. See response to 6.
6. Figure 2D and E: There are no statistics shown.
Response: Probably, reviewer means figure 3D. We believe that statistics is not relevant here as the full binding curves are shown. UV radiation would partially damage the protein but that is not expected to completely eliminate the binding of antibodies to their epitopes as spike oligomer is stabilised by several intra- and inter-protomer interactions. In fact, the modest reduction in binding is a sign of protein damage which is likely to cause substantial reduction in infectivity of the virus. Virus infection is complex process and required large-scale conformational transition which might be affected by UV-induced protein damage. As the reduction in binding (ELISA) is minor and there are only two comparators we believe it is better to show full inhibition curve rather than using single antibody concentration and recording binding observation multiple times for statistical difference. Rather, we performed ELISA in dose-dependent manner three times and given the representative results in the manuscript.
Reviewer response: The authors need to clarify what they are trying to say in figure 3D and E. It seems to this reviewer that the difference in figure 3 D right panel could be simply a difference of starting concentration, and not actually difference in biding. It does not look like there was any math done on the binding curves which can actually demonstrate that their claim is correct. This needs to be added by either providing statistics or at least showing if the K50 or equivalent is in fact different between the conditions.
7. There is a methodical concern in how the authors measure infectious virus samples. Obtaining infectious virus from gelatin filters should not be a problem. It is highly unusual to measure virus by qRT after inoculating cells after 48h. If the authors would try to actually perform plaque assays, they may be more successful in recovering virus. At least they should describe the limit of detection of the method they use and provide more details how this method is used to arrive at pfu/do comparisons between values. The conclusion drawn in L203, therefore, may be wrong, and the inability to detect virus is a result of a too high assay detection limit. If the authors insist on PCR, they should at least look at sgRNA, instead of gRNA, which would more readily support virus replication. Showing gRNA, without showing the same analysis pre-incubation of the cells, could simply mean there is viral RNA present, but not infectious virus. https://www.ncbi.nlm.nih.gov/pmc/articles/PMC7121658/#!po=25.0000 aerosol stability of SARS-CoV[1]2 https://www.ncbi.nlm.nih.gov/pmc/articles/PMC9413705/ gel filters able to capture virus from infected animals
Response: We believe there is some confusion here. Indeed, as reviewer pointed out, we did recover infectious virus particles from the gelatin membrane. It is not unusual to use rt-PCR assay and several published papers used it. This assay is simple and more sensitive than plaque forming unit (pfu) assay. There are different methods to quantify the virus; and no method provides absolutely real estimate of the infectious particles. Pfu assay in our setting do not form plaques when initial infectious titer is very low, although it works well when titers are moderate. Therefore, we chose to infect the highly receptive cells (vero-TMPRSS2 in this case) to amplify the virus and detect it along with all controls. In our experience, the method gives consistent results and is more sensitive than pfu assay. We are sure that the results mentioned in the said section are accurate to the best of our knowledge.
Reviewer response: It appears that I was not clear enough: Multiple other studies have demonstrated the ability to recover actual infectious virus from aerosolized virus on gel filters, which the authors claim they were not able to do. This suggests, as virus concentrations were rather high in their aerosol, that there may be a methodological problem at play. Checking for gRNA after such short time of incubation on cells has two problems: 1) gRNA is only measuring virus presence but not actually showing replication, and therefore not an indicator for infectious virus. 2) By limiting the time window such, the authors may be introducing a limit of detection which is too high. This then, in turn, pulls their data in question, as they may simply be missing infectious virus particles in their samples across experiments. I strongly urge that this data is verified using an actual TCID50 assay or a plaque assay. It is not standard in the field to use qRT PCR for gRNA for SARS-CoV-2 after infection of cells to determine the presence of infectious virus. If the authors do not provide such data, they will need to thoroughly discuss why they chose such an assay and the inherent limitations of detection. They also need to discuss why they are not able to recover virus from gel filters after aerosolization, while other groups have been able to do so with far lower virus concentration in the air. Using stock virus at a very high concentration to demonstrate that gRNA can be recovered from filters does not seem adequate proof, if it is in fact the limit of detection of the assay used which is the key problem. If the authors insist on this method, they will need to show an increase of gRNA after infections of the cells to rule out that it is not simply the same amount already present in the sample when inoculating. It is possible I missed this, but the qRT protocol should be referenced to clearly state if they are measuring gRNA or sgRNA, which would be helpful to make their case.
Author Response
The word file of responses to the comments is uploaded

Round 3
Reviewer 3 Report
I thank the authors for their responses. A response to minor (4) is missing.
Author Response
We have addressed the comments in the revised version.